

# Growing up is hard to do: a demographic model of survival and growth of Caribbean octocoral recruits

Howard R. Lasker[1,2] and Ángela Martínez-Quintana[2]

[1] Department of Geology, University at Buffalo, State University of New York, Buffalo, New York, United States of America
[2] Department of Environment and Sustainability, University at Buffalo, Buffalo, New York, United States of America

Corresponding author
Howard R. Lasker,
hlasker@buffalo.edu

## ABSTRACT

**Background:** Among species with size structured demography, population structure is determined by size specific survival and growth rates. This interplay is particularly important among recently settled colonial invertebrates for which survival is low and growth is the only way of escaping the high mortality that small colonies are subject to. Gorgonian corals settling on reefs can grow into colonies of millions of polyps and can be meters tall. However, all colonies start their benthic lives as single polyps, which are subject to high mortality rates. Annual survival among these species increases with size, reflecting the ability of colonies to increasingly survive partial mortality as they grow larger.

**Methods:** Data on survival and growth of gorgonian recruits in the genera *Eunicea* and *Pseudoplexaura* at two sites on the southern coast of St John, US Virgin Islands were used to generate a stage structured model that characterizes growth of recruits from 0.3 cm until they reach 5 cm height. The model used the frequency distributions of colony growth rates to incorporate variability into the model.

**Results:** High probabilities of zero and negative growth increase the time necessary to reach 5 cm and extends the demographic bottleneck caused by high mortality to multiple years. Only 5% of the recruits in the model survived and reached 5 cm height and, on average, recruits required 3 y to reach 5 cm height. Field measurements of recruitment rates often use colony height to differentiate recruits from older colonies, but height cannot unambiguously identify recruits due to the highly variable nature of colony growth. Our model shows how recruitment rates based on height average recruitment and survival across more than a single year, but size-based definitions of recruitment if consistently used can characterize the role of supply and early survival in the population dynamics of species.

## INTRODUCTION

The supply and initial survival of the youngest individuals entering a population are often key factors modulating the size of populations (*Varley & Gradwell, 1960*; *Gaines & Roughgarden, 1985*; *Caley et al., 1996*; *Steele, 1997*; *Tilman, 1997*). This may be especially

important among plant and invertebrate species where very large numbers of propagules/ offspring are produced, but relatively few survive the first months of life. The outcome of the supply and survival of individuals to a population is generally referred to as recruitment (*Caley et al., 1996*). For many species, the high mortality rates experienced at this stage in the life cycle creates a demographic bottleneck (*Yoshioka, 1996*; *Vermeij & Sandin, 2008*; *Gonzalez-Varo, Nora & Aparicio, 2012*; *Chong-Seng, Graham & Pratchett, 2014*; *Zimmer et al., 2014*; *Doropoulos et al., 2016*; *Sarribouette et al., 2022*). The scale of this bottleneck may be particularly great for modular organisms, such as colonial invertebrates and plants, where survival and growth are size-dependent and partial mortality can retard the rate at which individuals move through the life table (*Harper & White, 1974*; *Hughes, 1989*). Thus, quantifying recruitment is critical to understanding species' population dynamics, and to assessing species' resilience in the face of environmental change.

Although comparing annual recruitment rates among species is critically important, comparisons between studies are difficult due to differences in definitions of recruits and varying methodologies (*Jenkins, Marshall & Fraschetti, 2009*; *Pineda, Reyns & Starczak, 2009*). For instance, measures of recruitment are sensitive to the timing of measurement, as the supply of larvae or propagules to a population is seldom uniform over time and mortality rapidly reworks the numbers of recruits present as a cohort ages (c.f. *Pineda, Starczak & Stueckle, 2006*). In addition, the age of colonial invertebrates and plants is not always apparent from their size (*Hughes & Jackson, 1980*). In this study, we explore how censuses of populations that identify recruits on the basis of size alone may compound the settlement and survival of multiple cohorts, and assess how the use of a size-based definition of recruitment affects the measurement of recruit abundance and survival, using data for a suite of Caribbean octocoral species.

Branching Alcyonacean octocorals (colloquially called gorgonians) increasingly dominate the benthic macrofauna of many Caribbean coral reefs (*Ruzicka et al., 2013*; *Lenz et al., 2015*; *Tsounis & Edmunds, 2017*; *Lasker et al., 2020a*). Like many benthic invertebrates, they have bipartite life histories in which individuals disperse as larvae, which settle to the substratum and metamorphose into a polyp. These recruits develop into colonies as more polyps are generated *via* budding, producing an initial branch, which in most species then gives rise to additional branches (*Lasker & Sanchez, 2002*). The pattern of growth and branching, generates forms that include single branches, feather like plumes, bushy colonies with dichotomous branches, as well as, colonies whose branches anastomose to form large fans (*Lasker & Sanchez, 2002*). These forms all start their benthic lives as single polyps, which we refer to as primary polyps. Primary polyps are unambiguously recruits and some species can start budding new polyps within weeks of metamorphosis (*Tonra, Wells & Lasker, 2021*), but some may take months (*Hwang & Song, 2007*; *Sun, Hamel & Mercier, 2011*). Most studies of recruitment among octocorals have sampled on scales greater than the time needed for many recruits to bud, and therefore have included small single branch colonies as recruits, *i.e.*, those of a few centimeters in height (*Lasker, 2013*; *Lasker & Porto-Hannes, 2015*; *Martínez-Quintana & Lasker, 2021*; *Yoshioka, 1996*). *Martínez-Quintana & Lasker (2021)* distinguished primary

polyps or single-polyp recruits from colonial recruits. Mortality of single polyps and among the smallest colonies is considerable, but decreases as colonies grow (*Lasker & Porto Hannes, 2021*; *Martínez-Quintana & Lasker, 2021*; *Yoshioka, 1994*). Partial mortality, generally due to the actions of grazers, slows the growth of young colonies, extending the exposure of small colonies to high mortality rates and at the very least, extending the time needed to reach maturity (*Martínez-Quintana & Lasker, 2021*; *Sarribouette et al., 2022*).

For most benthic invertebrates, recruitment rates have been measured using variants of two methodologies, the deployment and collection of settlement tiles, or *in situ* censuses of recruits on natural or artificial substrata (c.f. *Martínez-Quintana & Lasker, 2021*). The use of settlement tiles has the advantage that tiles are deployed for a set interval. Thus, all of the individuals on the tiles are unambiguously recruits of similar age. Censuses of *in situ* populations, while characterizing recruitment in a more natural setting, also require the investigator to distinguish recruits from small but older colonies. Single-polyp recruits persist in that state for weeks to months before dividing and becoming colonial (*Tonra, Wells & Lasker, 2021*). Only censuses in which recruits are mapped, or those performed at short intervals can unambiguously distinguish recruits from small juveniles, *i.e.*, small and not reproductive colonies that are older than 1 year. Similar issues can be found in the dynamics of plant populations. For instance, seedlings of similar sizes may have germinated from seeds produced in different years (*Wang et al., 2013*). Similarly, small understory trees can differ in age by years (*Clark & Clark, 2001*). In the case of octocorals, we have used colony height to differentiate recruits, but that is sensitive to intra and inter species-specific growth rates. While the presence of biases introduced by differing sampling schemes is generally acknowledged, its effects on the recruitment of branching marine invertebrates as octocorals are seldom quantified.

In this study, we characterize the probability of octocoral recruits ever reaching juvenile size and address the difficulty of classifying as recruits organisms with a size-dependent life history. To determine growth and survival rates, we followed the fate of colonial recruits. Next, we construct a stage structured model to quantify the duration of the recruitment bottleneck, and the percentage of recruits that survive and grow to the size at which they can be considered juveniles. Finally, we consider how our understanding of the recruitment bottleneck is shaped by the definition of recruit. Our goals are two-fold: to understand the population dynamics of these small and young colonies, and to assess the effects of including small but older colonies as recruits in estimates of recruitment rate. We address the questions of the likelihood of a colony reaching a size at which it would no longer be considered a recruit, and the proportion of colonies that would mistakenly be characterized as recruits.

## MATERIALS AND METHODS

### Field data

Growth rates and survival of small octocorals ranging in size from 3 mm to 5 cm height were obtained from observations of colonies on two reefs on the southern shore of St John, US Virgin Islands, Grootpan Bay (18° 18.360′N, 64° 43.140′W) and Europa Bay (18°

19.016′N, 64° 43.798′W). Study sites and octocoral communities have been described in detail (*Tsounis et al., 2018*). Briefly, both sites are fringing reefs at 5–9 m depth, within 100 m of the coastline on the southern shore of St John, within the US Virgin Is. National Park. The reef substratum is a mix of carbonate rock, primarily originated from dead scleractinian corals, igneous rock and unconsolidated carbonate sediments. Scleractinian corals, octocorals, algal turfs, macroalgae (most commonly *Dictyota* spp.), and sponges are abundant benthic taxa. Crustose coralline algae and the calcareous alga *Ramicrusta* sp. are also common on hard substrata. Water movements at both sites are dominated by wave driven oscillations. Previous studies have identified the site at Grootpan Bay as East Cabritte (*Lasker et al., 2020b*), but we now use the designation that best fits nautical charts of the area. The research was conducted under National Park Service Permits VIIS-2016-SCI-0011, VIIS-2017-SCI-0010, VIIS-2018-SCI-0011, and VIIS-2019-SCI-0011.

The methods used to characterize recruit survival and growth are detailed in *Martínez-Quintana & Lasker (2021)* and summarized as follows. At the two sites, a total of 359 *Eunicea* and *Pseudoplexaura* spp. recruits, defined as colonies ≤5 cm height (*Lasker et al., 2020b*), were arbitrarily selected and mapped along two 10 m transects at Grootpan Bay (192 recruits), and three 10 m transects at Europa Bay (167 recruits). Species level identifications of the colonies would have required destructive sampling, and thus were not possible, but multiple years of observing recruits and their subsequent growth gives us confidence that we could distinguish genera. We measured colony heights to the nearest mm in August 2016. Recruits were mapped by recording their distance to fixed bolts in the reef, and distances between a recruit and each pair of bolts were used to locate each recruit. Measurements of surviving individuals were repeated 6, 11, 15, 20, 23, 31 and 36 months later. Although present, recently settled single polyps were not mapped. Studies of settlers on tiles have shown that single-polyp recruits suffer very high mortality rates (*Evans, Coffroth & Lasker, 2013*; *Lasker, Kim & Coffroth, 1998*; *Lasker & Porto Hannes, 2021*; *Martínez-Quintana & Lasker, 2021*; *Wells et al., 2021*), and it was impractical to map and monitor the hundreds of single polyp recruits that would have been needed to characterize their survival and growth. Thus, our analyses are based on the fates of colonies that had already survived the first months following settlement.

## Analysis and statistical modelling

The mapped recruits were placed in size (height) classes of 0.3–0.5 cm, 0.6–1.1 cm, 1.2–2.0 cm, and 2.1–5.0 cm. The size classes follow those used in *Martínez-Quintana & Lasker (2021)*. Those size class boundaries were established taking into account the size structure of the recruit population at our sites and based on the 25th, 50th and 75th percentiles of the recruits. Recruits from the two sites and taxa were pooled, as earlier analyses had shown there were no significant differences in their growth rates and survivorship (*Martínez-Quintana & Lasker, 2021*). Transition matrices, the matrix of probabilities of a recruit in a given size class surviving, and either remaining in that size class or transitioning to a larger (or smaller) size class were calculated in two ways. In the first approach, which we refer to as the "empirical model", the fate of each surviving colony in each interval was scored according to whether it remained in its size class, increased, or decreased in size to one of

the other size classes. Probabilities of each transition were calculated as the ratio between the number of individuals in each transition and the number of recruits in the size class. For logistical reasons, time intervals between censuses were not all 6 months, and only data from 6-month intervals were used to assess the probability of changing into a different size class.

Our second approach, the "growth model", used the observed growth rates of colonies of different sizes in conjunction with variance of those growth rates to calculate size-specific probabilities of changes in height great enough to place a colony in a different size class. Developing transition matrices from growth and survival observed in a few time intervals can be especially useful when it is impractical to follow individuals through the entire life table. In this approach, we partitioned the recruits into instances of growth, no-growth, or negative growth (*i.e.*, tissue loss, most likely due to grazing), and calculated changes in height within each growth class and the proportion of individuals in each growth class that changed in height. The positive and negative growth rates for each of the four size classes were tested for their fit to normal, log-normal, gamma and Weibull distributions. Distributions of the negative growth rates were determined using the absolute values of the growth rates. Zero growth rates were not included in fitting distributions. Distribution fitting was conducted with the Excel add-in Xrealstats, (Zaiontz, C., Real Statistics Using Excel. www.real-statistics.com).

Transition probabilities were calculated as depicted in Fig. 1. First, the proportion of recruits of a given size were determined. Next, the likelihood of a recruit in a given size class increasing, decreasing, or remaining in the same size was calculated based on the observed proportion of recruits that had a positive, negative, or no growth rate. Then, given an initial size, the probability that the growth would be sufficient to lead to a change in size class was determined from the distribution of growth rates. Those probabilities were calculated for recruits of every size within each class (at 0.1 cm intervals) and the probabilities were averaged creating a single probability for each size class. A final >5 cm size class was used to keep track of recruits that reached 5 cm height and that were no longer subject to mortality or shrinkage.

The process described above generated two sets of transition matrices, the first was based simply on the observed transition probabilities, *i.e.*, the "empirical model", and the second transition matrix was based on the growth data, *i.e.*, the "growth model". The empirical model has the advantage of being direct measurements of transition probabilities, but the disadvantages of calculating probabilities based on relatively few individuals, which were not distributed uniformly within the four size classes. The growth model also suffers from relatively few observations in some categories, but has the advantage of incorporating variance in growth rates in the analysis.

The dynamics of a cohort of recruits was modeled using the transition matrix as a simple stage-structured model (*Caswell, 2001*), with the number of recruits in the size class calculated as the product of the transition probabilities, and the vector of numbers of recruits in each size class in the preceding time step. We used the models in two fashions, first, seeding a population with an initial cohort of 0.3–0.5 cm recruits, and then following the fate of the cohort. The model was initiated with 100 settlers, which were followed
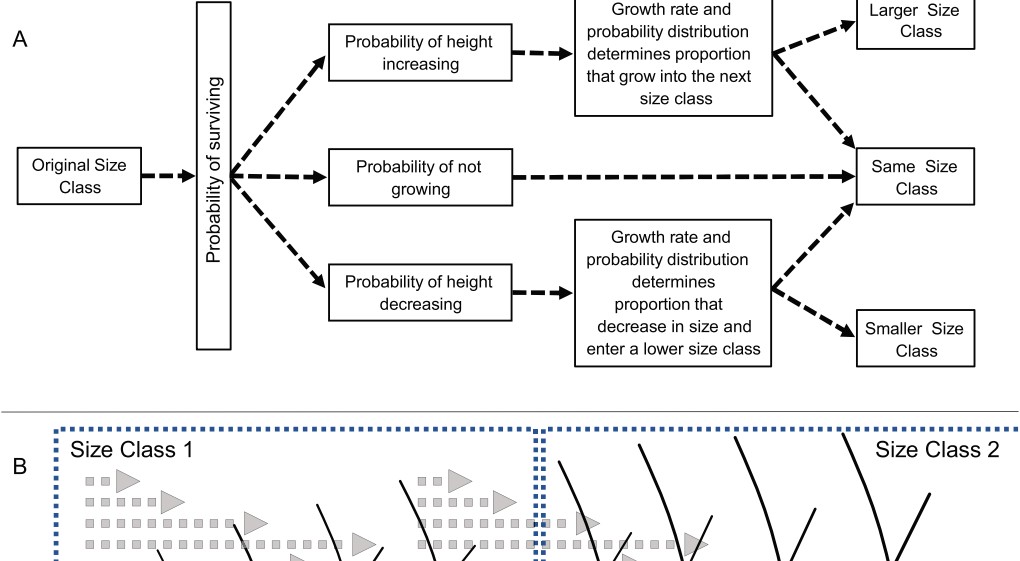

**Figure 1 Diagram of the modeling process.** (A) Schematic diagram of the process used to calculate transition probabilities for recruits of different sizes. (B) Illustration of how a colony's size, growth rate and variance in growth rates determines the likelihood of a colony growing, and if growth is great enough entering the next larger size class.

through 10 years. In particular, we were interested in the proportion of recruits that survived and reached the >5 cm size class, *i.e.*, survived the recruit phase, and what was the mean time to reach 5 cm height. A second analysis used a constant input of newly settled recruits in order to characterize the size distribution of recruits under equilibrium conditions.

Sensitivity and elasticity of the proportion of recruits reaching 5 cm height and the time required to do so were conducted by changing either the transition probabilities of the empirical transition matrix or by altering differing combinations of growth rates, probabilities of having negative growth, and survival for the growth model. Sensitivity analysis of the growth model also allowed us to examine the effects of changes to growth rates, and the probabilities of undergoing partial mortality and overall mortality, all of which are less abstract than transition probabilities. Elasticity characterizes the effects of changing probabilities as the proportionate change relative to the original values. In this case, evaluations of sensitivity and elasticity based on the eigen vectors of the transition matrix are inappropriate, since the transition matrices do not include probabilities for the addition or the absent of new settlers, and the equilibrium outcome is a population made up of a small number of 5 cm colonies. We calculated sensitivity and elasticity to changes in each transition probability for the proportion of colonies growing out of the recruit size class within 5 years, and for the average time recruits required to reach that size. Elasticities were also calculated for changes in growth rates, the probability of suffering partial mortality (*i.e.*, negative growth) and survivorship. The empirical and growth models were

**Table 1 Transition matrix of the empirical model of recruit growth and survival.**

| Size class | 0–0.5 cm | 0.6–1.1 cm | 1.2–2 cm | 2.1–5 cm | >5 cm |
|---|---|---|---|---|---|
| 0–0.5 cm | 0.198 | 0.054 | 0.011 | 0.019 | 0 |
| 0.6–1.1 cm | 0.208 | 0.240 | 0.126 | 0.077 | 0 |
| 1.2–2 cm | 0.050 | 0.109 | 0.161 | 0.038 | 0 |
| 2.1–5 cm | 0.040 | 0.098 | 0.230 | 0.442 | 0 |
| >5 cm | 0 | 0 | 0 | 0.077 | 1 |

both tested by comparing the predicted size class distribution of the recruits to observed abundances of recruits previously reported by *Martínez-Quintana & Lasker (2021)*.

## RESULTS

A general characterization of the survival and growth of the recruits used in the analysis has been presented in *Martínez-Quintana & Lasker (2021)*. The empirical transition matrix is depicted in Table 1. Data for the transition matrix were limited to cases in which the interval between observations was 6 months, which reduced the data set to 302 observations. Out of those recruits, 156 died, and the remaining 142 were used to determine transition probabilities. The number of recruits predicted to reach 5 cm height over a 10 y time span and the predicted size frequency distribution of recruits are presented in Fig. 2. In the simulation, almost all (99%) the recruits were under 5 cm after 1 y. The number of recruits reaching 5 cm approached an asymptote at approximately 5 y, at which only 1.65% of the initial recruit cohort reached 5 cm height. The proportion reaching 5 cm height was marginally greater at 10 y (1.67%). Among the recruits reaching 5 cm in 5 y, the mean time required to grow to 5 cm was 2.70 y. Restated, after 1 y, 99% of the surviving recruits were less than 5 cm height, after 5 y 98% of the cohort died and most (90%) of the surviving recruits reached 5 cm height.

In the simulation with the empirical model, we found the average age of recruits present in the population was dependent on the time at which the population was surveyed. When recruits were added each year, and then surveyed at the time of the next recruitment event, the combination of newly introduced recruits with older recruits that had not yet grown past 5 cm height led to recruits being in average 1.4 y old. Whereas, when surveying recruits 6 months after settlement, the predicted average age of colonies <5 cm height, was 0.4 y. Our simulations predicted recruits <0.6 cm to be less abundant than the 0.6–1.1 cm size class, which was significantly different from the observed distribution (Fig. 3; Chi-square analysis, $p = 5.8 \times 10^{-8}$). The sensitivity and elasticity of the empirical model are presented in Table S1.

The growth model was based on 707 observations in which growth or mortality could be determined. Of those, 434 observations were of recruits that survived at least a one time-interval and for which a growth rate could be calculated (Fig. 4). In general, growth rates were variable. The distributions of the growth rates are truncated because there cannot be negative growth rates that would take the largest colony in the class to 0.0 cm. The ranges of the growth rates for the <0.6, 0.6–1.0, and 1.1–2 cm size classes were similar,

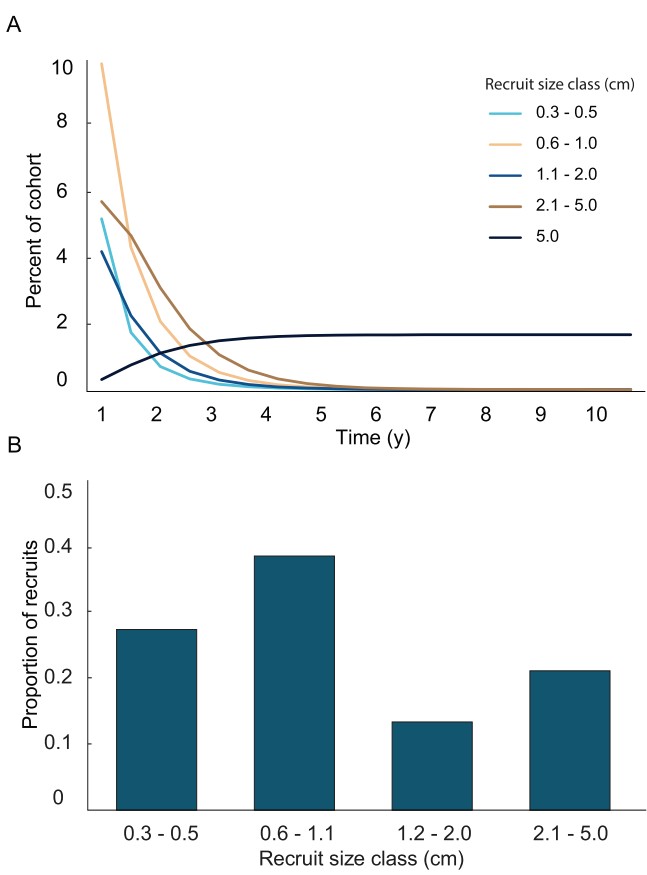

**Figure 2 Results from the empirical recruit matrix model.** (A) Proportion of recruits expected to reach 5 cm height over time based on the transition matrix derived from observations of the recruits mapped at Grootpan and Europa Bays, St John (USVI). (B) Size distribution of recruits after 10 y assuming constant settlement each year.

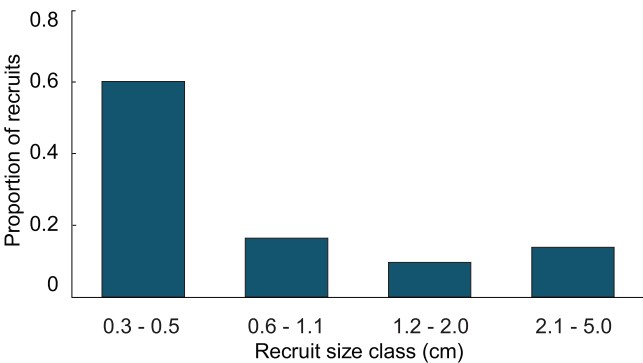

**Figure 3 Size class distribution of recruits found on natural substratum on reefs at Grootpan and Europa Bays, St John.**

while the range for the larger 2.1–5 cm size class was over twice as great. The distribution of positive growth rates has longer tails. Size class specific mortality rates were previously calculated using a Cox proportional hazards model reported in *Martínez-Quintana &*

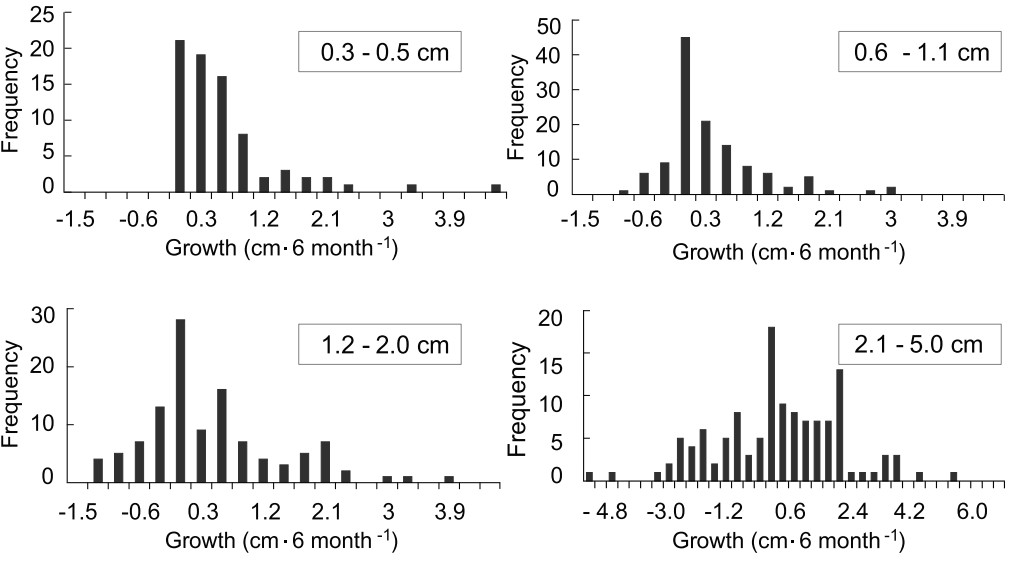

**Figure 4 Frequency distributions of growth rates of mapped octocoral recruits monitored at approximately 6-month intervals on the south shore of St. John, US Virgin Islands.**

*Lasker (2021)* and those values were used in the growth model to establish the proportion of recruits in each class that survived.

Variation in growth rates was incorporated into the growth model by fitting the positive and negative growth rates of each size class to probability distributions. Then, the calculated distributions were used to determine the probability a recruit of a given size exhibiting a large enough change in height to transition into one of the other size classes. No single type of distribution fit the positive and negative growth rates in all four size classes. The best fits were the log-normal distributions. Only three out of the eight distributions were significantly different from a log-normal distribution, and they did not fit any of the other tested distributions. The log-normal distributions were used to calculate transition probabilities for each of the size classes, and the transition matrix (Table 1), then they were used to calculate the dynamics of a cohort (Fig. 5A). The predicted proportion of recruits reaching 5 cm height within 5 years (5.09%) was greater in the growth model than in the empirical model. 4.3% of the initial cohort reached 5 cm in 3 y, which is comparable to our observations in the field, *i.e.*, out of 359 mapped colonies, 2.2% reached 5 cm in 3 y. The predicted average time to reach 5 cm in height, 2.95 y, was similar between models. After 5 years of adding recruits to the population, the average age of colonies <5 cm was 1.52 y (Fig. 5B). Unlike the empirical model, the growth model predicted that abundances of recruits within the 0.3–0.5 cm size class would be greater than either of the other size classes. The size class distribution of recruits predicted in the growth model was not significantly different from the observed size class distributions on the reef (Fig. 6, Chi-square test, $p = 0.14$).

The importance of rapidly moving through the size class distribution, *i.e.*, growing fast, is evident in the sensitivity and elasticity analyses (Table 2). In the sensitivity analysis, increasing the probability of transition to a larger size class had the greatest effect on the

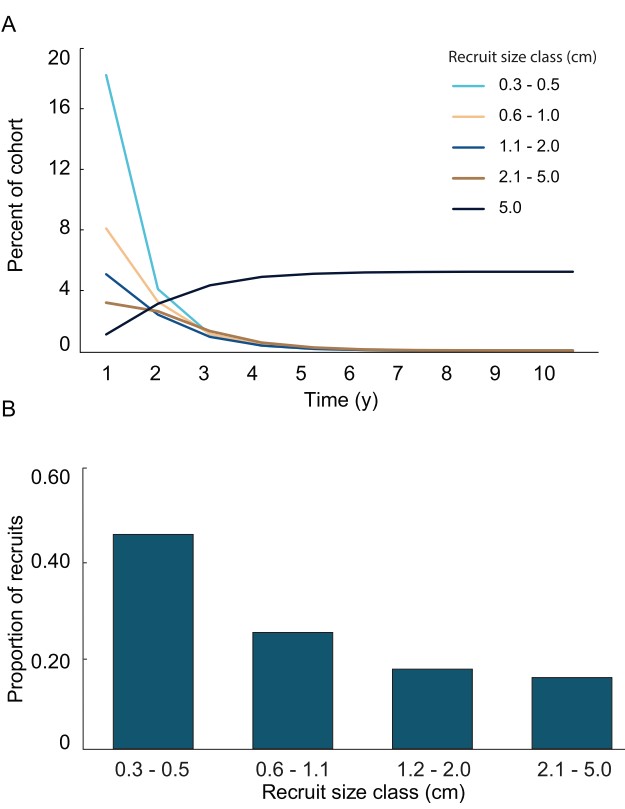

**Figure 5 Results from the growth model of recruit demographics.** (A) Proportion of recruits expected to reach 5 cm height over time based on the transition matrix derived from survival and growth rates of the recruits mapped at Grootpan and Europa Bays, St John. (B) Size distribution of recruits after 10 y assuming constant settlement each year.

proportion of recruits surviving and growing to 5 cm. The sensitivity and elasticity of the transition probabilities associated with growth out of the smallest size class were higher than those involving larger colonies. In the elasticity analyses, increasing the probability of transition to a juvenile stage had a lesser effect on surviving and growing to 5 cm, but the conclusions of both analyses are similar in that growing from the smallest size class and to the >5 cm class have the greatest impact on the proportion of recruits reaching the juvenile size class, and on the time required to grow to that size. Interestingly, all of the sensitivity values for the proportion of recruits surviving to ≥5 cm were positive, including those for decreasing size, *i.e.*, those above the diagonal in the matrix of transition probabilities in Table 2. This effect occurred because increases in any single transition probability, even if it affects colony shrinkage, is an increase in survival for the colonies in the size class.

Increasing transition probabilities in the growth model slightly reduced the time at which surviving colonies reached 5 cm height. Fig. 6 depicts the effects of altering survival, growth rate, and probabilities of suffering partial mortality on the probability of a recruit surviving and growing to 5 cm height and the time to reach that size. The changes were applied uniformly to all size classes in these analyses. The slopes depicted in Fig. 6 are analogous to elasticity, and changes to survival had by far the greatest effects on the

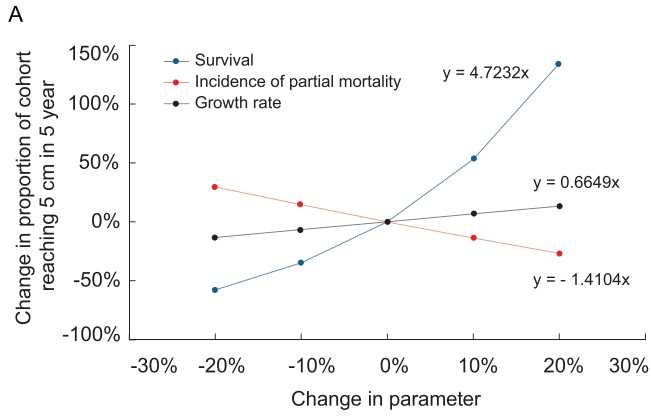

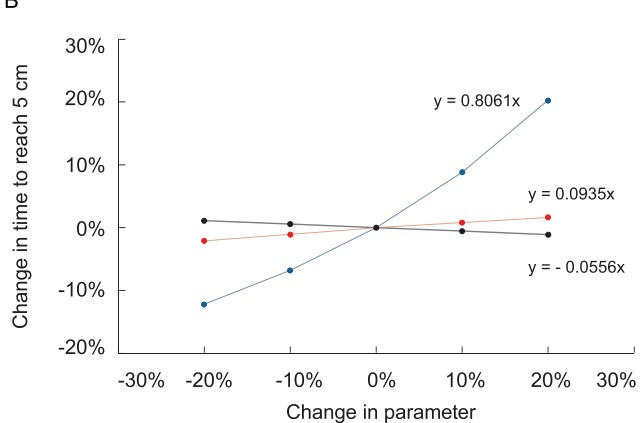

**Figure 6** **Effects of increasing survival, growth rate and the proportion of colonies suffering partial mortality on the proportion of recruits surviving and reaching 5 cm height and the average time to reach 5 cm height.** Slopes of the linear regressions are analogous to elasticity.

probability of a recruit surviving and growing to 5 cm height and on the average time required to reach that size.

## DISCUSSION

Among many plant species and benthic invertebrates, size or developmental stage, not age, defines probabilities of survival, growth and reproduction (*Harper & White, 1974*; *Hughes & Jackson, 1980*; *Jackson, 1985*). Among clonal species, partial mortality, the loss of tissue from a colony or plant, can reduce the size of an individual, increasing the time they remain in vulnerable size classes and affecting its survival and fecundity (*Harper & White, 1974*, *Hughes & Jackson, 1980*). As illustrated in the simulation of octocoral recruit growth and survival, this is particularly consequential among colonial benthic invertebrates. The simulations show that variable growth rates and partial mortality slowed the rate at which recruits became juveniles, *i.e.*, colonies >5 cm height that are not reproductive. The onset of sexual reproduction usually occurs at heights of 15–20 cm (*Kahng, Benayahu & Lasker, 2011*). On average, our simulations show recruits need ~3 y for recruits to grow to 5 cm. Perhaps more importantly, exposure to high mortality rates experienced by colonies that remained small over multiple years reduced the proportion of recruits that

**Table 2 Transition matrix, sensitivity, and elasticity of the growth model of recruit growth and survival.**

| | Size class | | | | |
|---|---|---|---|---|---|
| Size class | 0.3–0.5 cm | 0.6–1.1 cm | 1.2–2 cm | 2.1–5 cm | >5 cm |
| 0.3–0.5 | 0.417 | 0.072 | 0.024 | 0.031 | 0.0 |
| 0.6–1.1 | 0.095 | 0.374 | 0.099 | 0.024 | 0.0 |
| 1.2–2.0 | 0.054 | 0.084 | 0.356 | 0.059 | 0.0 |
| 2.1–4.9 | 0.019 | 0.076 | 0.146 | 0.469 | 0.0 |
| ≥5.0 | 0.002 | 0.016 | 0.051 | 0.207 | 1.0 |

Proportion of colonies attaining 5 cm height

Sensitivity

| Size class | 0.3–0.5 | 0.6–1.1 | 1.2–2.0 | 2.1–5.0 |
|---|---|---|---|---|
| 0.3–0.5 | 8.63 | 1.28 | 19.14 | 18.88 |
| 0.6–1.1 | 18.28 | 2.92 | 20.17 | 19.67 |
| 1.2–2.0 | 33.09 | 5.36 | 21.86 | 20.92 |
| 2.1–4.9 | 91.63 | 12.17 | 26.37 | 24.58 |
| ≥5.0 | 176.61 | 30.34 | 38.69 | 34.56 |

Elasticity

| Size class | 0.3–0.5 | 0.6–1.1 | 1.2–2.0 | 2.1–5.0 |
|---|---|---|---|---|
| 0.6–1.1 | 1.50 | 1.76 | 1.65 | 1.61 |
| 1.2–2.0 | 2.71 | 1.98 | 1.79 | 1.71 |
| 2.1–4.9 | 7.50 | 2.59 | 2.16 | 2.01 |
| ≥ 5.0 | 14.46 | 4.22 | 3.17 | 2.83 |

Time (years) to reach 5 cm

Sensitivity

| Size class | 0.3–0.5 | 0.6–1.1 | 1.2–2.0 | 2.1–5.0 |
|---|---|---|---|---|
| 0.6–1.1 | −19.01 | −18.17 | −18.41 | −18.27 |
| 1.2–2.0 | −19.56 | −18.04 | −18.23 | −17.96 |
| 2.1–4.9 | −20.48 | −18.27 | −18.24 | −17.51 |
| ≥5.0 | −21.04 | −20.08 | −19.49 | −18.80 |

Elasticity

| Size class | 0.3–0.5 | 0.6–1.1 | 1.2–2.0 | 2.1–5.0 |
|---|---|---|---|---|
| 0.6–1.1 | −2.51 | −2.40 | −2.43 | −2.42 |
| 1.2–2.0 | −2.59 | −2.39 | −2.41 | −2.37 |
| 2.1–4.9 | −2.71 | −2.42 | −2.41 | −2.32 |
| ≥5.0 | −2.78 | −2.66 | −2.58 | −2.49 |

survived to the juvenile stage, and only 5.1% of the cohort reached 5 cm. There are relatively few comparable studies for other species, but those available suggest differences across taxa and/or regions. *Lasker (1990)*, also using a size-dependent matrix model, estimated that 13% of *Plexaura kuna* colonies in a 0–9.9 cm size class at a site in Panama would eventually survive and grow out of that size class. Those data were largely based on survival and growth of colonies that had developed from fragments, and their survival would have been greater than most of the colonies in the 0.3–5 cm size class that we report

on. Thus, greater success would be expected. In contrast to the St John and Panama models, *Yoshioka (1994)* generated a size structured model of gorgonian dynamics at a site in Puerto Rico in which 54% of <6 cm colonies grow out of that recruit class in a single year. The Puerto Rico site was dominated by *Antillogorgia* spp., which are fast growing species (*Yoshioka & Yoshioka, 1991*). *Gotelli (1991)* developed a similar model for *Leptogorgia virgulata* which is not a reef species, and reported 16% of recruits between 0.5–4 cm grew out of the recruit size class within a single month, which extrapolated over the course of a year suggests 43% of the recruits will grow to 4 cm.

The low percentage of recruits that grew to 5 cm in the simulations of St John octocoral recruits was in large part due to repeated exposure to annual mortality rates, which ranged from 21% to 41%, depending on colony size. Those rates compounded over multiple years led the growth model to predict 95% mortality of the recruits over 5 y, with most of that mortality occurring in the first few years. Furthermore, the low probability of reaching the juvenile size class is likely an overestimate because the data used to parameterize the simulations, and the simulations themselves, started with recruits within a 0.3–0.5 cm size class, which excluded newly settled, single polyp recruits. Previous work at Europa and Grootpan Bays found that newly metamorphosed *Plexaura homomalla* polyps that settled on unglazed stoneware clay tiles suffered 87% mortality over 59 d (*Martínez-Quintana & Lasker, 2021*). *Tonra, Wells & Lasker (2021)* observed *P. homomalla* recruits reached three polyps (2–3 mm) in 10 weeks. If we assume recruits within the 0.3–0.5 cm size class were at least 1–2 months old, adding 1–2 months mortality would reduce our estimates of survival to the juvenile stage an additional 65–87%.

In our simulations, negative growth rates and probabilities of partial mortality led colonies to remain in the recruit size class for over a year. The effects of these factors are evident in the sensitivity and electivity analyses of the growth model. Fig. 6 shows the effects are virtually reciprocal, with a given increase in growth rate compensating for a similar increase in partial mortality. Annual survival had by far the greatest effect on the amount of time to reach 5 cm and the likelihood of a colony escaping, *i.e.*, growing out of, the 5 cm recruit class.

High mortality among newly settled individuals, and the small colonies they develop into, occurs across a range of benthic taxa (*Babcock & Mundy, 1996*; *Bramanti et al., 2005*; *Gomez et al., 2014*; *Keough & Downes, 1982*; *Linares et al., 2008*; *Martínez-Quintana & Lasker, 2021*; *Miller, Weil & Szmant, 2000*), with estimates of 98% mortality in 30 d for the octocorals *Plexaura kuna*, (*Lasker, Kim & Coffroth, 1998*) and 99% for *Briareum asbestinum*, (*Evans, Coffroth & Lasker, 2013*). Much of the high mortality reported for single-polyps is probably attributable to predation. Gorgonian octocorals naturally settle on the undersides of tiles and in the field within holes and crevices (*Martínez-Quintana & Lasker, 2021*; *Wells et al., 2021*). Those microhabitats offer recruits some protection from predators and grazers. When fishes have access to the substratum, mortality is markedly higher, and field experiments that restricted access of fishes have led to reduced mortality of octocoral settlers (*Evans, Coffroth & Lasker, 2013*; *Lasker, Kim & Coffroth, 1998*). Mortality could also occur when competitors for space such as ascidians, sponges or other cnidarians and turf or macroalgae kill or overgrow adjacent recruits. *Wells et al. (2021,*

2022) examined how some of these processes affect single polyp recruits, concluding that while a high density of turf algae has a negative effect on the survival of single polyps, intermediate densities are associated with higher survival. They interpreted that pattern as indicative of the interplay between algal growth, which would inhibit recruit success, and grazing which would adversely affect both, recruits and algae. As reported here and in Yoshioka (1994), Gomez et al. (2014), and Martínez-Quintana & Lasker (2021), mortality rate declines with size which is probably a consequence of larger colonies exhibiting greater resilience from partial mortality, as well as their height enabling them to escape conditions on the substratum.

Our field observations along with our simulations illustrate that when 5 cm is used as the upper bound of a recruit class, many "recruits" will be older than 1 year. Although these colonies are not "recruits" in the sense of having been added to the population in a single year, distinguishing these colonies on the basis of their height regardless of their age is necessary for understanding the dynamics of the population, and for predicting the likelihood of colonies surviving and becoming juveniles. This is similar to plant ecologists characterizing the abundance of seeds in a seed bank regardless of their age (Wang et al., 2013), and similarly identifying saplings on the basis of size not age (c.f., Clark & Clark, 2001). The similarity of colonial recruits to seeds and saplings is particularly striking, as like the octocoral recruits discussed here, individuals can persist in those classes for highly variable lengths of time, and undergo highly variable and even negative growth.

The greatest improvement in the methodology used here would be to generate species-specific data. There were not genus specific differences in growth rates at our sites (Martínez-Quintana & Lasker, 2021), but Yoshioka & Yoshioka (1991) reported species-specific differences in growth at sites in Puerto Rico. Teasing apart net and potential growth rates and increasing sampling sizes may help to identify species-specific growth and survival strategies. Differentiating potential growth rates will require combinations of field experiments that remove grazers along with field measurements taken at frequencies that can differentiate grazing from slow growth. Although species level identification of recruits has been possible in some communities (Gomez et al., 2014), that is not yet the case for the Caribbean fauna. Among the species in this study, species level identification through visual inspection was only possible for a small number of colonies. Identifications from sclerites in the smallest colonies requires destructive sampling, and sclerites in recruits are often not fully formed and thus lack the characteristics used to distinguish adults (unpublished observations). Molecular markers have been identified for *Antillogorgia* spp. (Jamison & Lasker, 2008), but markers that work across a broader array of taxa are needed, and molecular identifications that do not require destructive sampling have not been developed.

## CONCLUSIONS

To better understand the population dynamics of recruits, we modeled survival and growth rates of octocoral recruits using size-structured models based on the survival and growth of recruits that had been followed for over 3 y. A simple model built around the

fates of recruits generated model predictions of the recruit size class distribution that did not fit the size class distribution at the study sites. However, using probability distributions of growth rates to incorporate variation in growth rates into the model led to a predicted size class distribution that was congruent with field data. Using 5 cm as the size differentiating recruits from juvenile colonies in our growth model, we found that 5% of recruits reached 5 cm height from a starting size of 0.3 cm and the median time to reach 5 cm height was 3 y.

Both the simulations and the underlying data on the mapped recruits demonstrated that only a fraction of the colonies that were <5 cm height were recruits, *i.e.*, were added to the population in a single year. Size-based definitions of recruits are arbitrary. Quantifying the number of settlers would provide an unambiguous characterization of recruitment and the closest we can get to settlement is defining recruits as single-polyps. That will require sampling schemes that incorporate differences among species in the timing of spawning/planulation, and differences in the time required for single-polyp recruits to develop into colonies.

The difficulty of generating a true annual recruitment rate, suggests size-based definitions of recruits are a necessary compromise in field measures of recruitment, with the upper bound set at or near the maximum size a newly established colony could reach in a year. As for all sampling schemes, such recruitment measures will miss individuals that die and grow more quickly or more slowly than expected. In the simulation presented in this study, 46% of the <5 cm colonies were over 1 y old, and on average recruits took 3 y to reach 5 cm. Thus, when 5 cm is used as an upper bound, the census of recruits will yield a 1–3 y time averaged estimate of recruitment. Restricting the definition of recruits in our simulation to 2 cm and below would lead to a recruit class in which 71% of the colonies were 1 year-old, but would miss 14% of the settlers, which due to higher growth rates grew to over 2 cm height within a year. To the extent that recruits can be identified to genera or species, the accuracy of size-based censuses for recruits could be improved with taxon-specific definitions of the size of the recruit class as those done by *Gomez et al. (2014)*, and with taxon-specific growth and survival data. Although size-based definitions of a recruit provide an imprecise, index of recruitment, if applied consistently over time and space, they can be used to assess the role of recruitment on community assemblages among localities and over time.

The dynamics of Caribbean octocoral populations have taken on increasing importance as octocorals have been more resilient to environmental stressors than scleractinians corals, and *Lasker et al. (2020a, 2020b)*, and *Martínez-Quintana & Lasker (2021)* have suggested that successful recruitment has been at the core of that resilience. More complete characterizations of recruit dynamics are needed, and models like the growth model we used can be an important tool, in further exploring the factors limiting recruit success. Conducting such studies for both octocorals and scleractinians will be important in understanding the differences in resilience of both taxa on contemporary reefs and how they are affected by climate change.

## ACKNOWLEDGEMENTS

Special thanks to Jacqueline Krawiecki, Christopher Wells and Kaitlyn Tonra for providing help and support in the field. Thanks to Peter Edmunds (California State University, Northridge) for logistical help in the field, and for helpful comments throughout the project. Christopher Wells and Mary Alice Coffroth along with the BURR lab at the University at Buffalo made helpful comments as we developed the manuscript. We also thank the staff of the University of the Virgin Islands, Virgin Islands Environmental Resource Station, for logistical support in the field.

### Funding

This research has been funded by grants from the US National Science Foundation to Howard R. Lasker (OCE1756381 and OCE1801475), and by funding awarded to Ángela Martínez-Quintana by the Mark Diamond Research Fund (Graduate Student Association at the University at Buffalo). The funders had no role in study design, data collection and analysis, decision to publish, or preparation of the manuscript.

### Grant Disclosures

The following grant information was disclosed by the authors:
US National Science Foundation: OCE1756381 and OCE1801475.
Mark Diamond Research Fund.

### Competing Interests

The authors declare that they have no competing interests.

### Author Contributions

- Howard R. Lasker conceived and designed the fieldwork and model, analyzed the data, prepared figures and/or tables, authored or reviewed drafts of the article, and approved the final draft.
- Ángela Martínez-Quintana conceived and designed the field work, conducted the field work, analyzed the data, prepared figures, authored and reviewed drafts of the article, and approved the final draft.

### Field Study Permissions

The following information was supplied relating to field study approvals (*i.e.*, approving body and any reference numbers):

The research was conducted under permits from the US National Park Service (VIIS-2016-SCI-0011,VIIS-2017-SCI-0010, VIIS-2018-SCI-0011, VIIS-2019-SCI-0011).

### Data Availability

Data on the abundance of recruits at Grootpan Bay are available at: Lasker, H. (2021) Octocoral recruitment surveys on transects at five sites on the south shore of St. John, US

Virgin Islands, 2014-2019. Biological and Chemical Oceanography Data Management Office (BCO-DMO). (Version 1) Version Date 2021-08-31 [if applicable, indicate subset used]. DOI 10.26008/1912/bco-dmo.851382.1

The data on the growth rates of mapped recruits and the matrix model are available in the Supplemental Files.

## Supplemental Information

Supplemental information for this article can be found online at http://dx.doi.org/10.7717/peerj.14386#supplemental-information.

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
