# Peer review of "Growing up is hard to do: a demographic model of survival and growth of Caribbean octocoral recruits"

_PeerJ, doi:10.7717/peerj.14386_

## Round 0.1 · original submission · Major Revisions

I have heard back from three reviewers, all of whom have offered constructive comments that should help you improve your work. In particular, I agree with two of the reviewers who have asked for more Discussion on the implications of your findings for Caribbean coral reefs. Even if brief, a few comments would be welcome.
I look forward to seeing a revised version of your paper.

Reviewer 1 ·

Basic reporting

The authors have reported survival and growth of important octocoral species in Caribbean, and probability of those recruits to grow and survive in a decade. While I agree that this information is essentially useful, especially for our understanding in the ecology of octocorals, I believe this manuscript needs to be improved in some areas. The structure of the manuscript is generally good, but I have some specific comments/suggestions that the authors might want to consider into. I have attached it in the additional comments.

Experimental design

The authors have reported survival and growth of octocoral recruits in Caribbean. Given the difficulty of tracking recruits survival and growth for long term periods, the authors conducted transition matrices of mapped recruits in order to provide insights as to the fate of the colonies through the years using the first few years quantified data. I believe the experimental design of this study is OK. Some of the data for this study has already been presented in Martinez-Quintana and Lasker et al. 2021, and the strength of this manuscript lies on the provision of transition matrices, sensitivity, and elasticity models of those data.

Validity of the findings

The findings of this study is okay. However, I have provided in the additional comments my specific comments to the authors to improve their manuscript.

Additional comments

Below are my detailed suggestions to improve the manuscript. All the best to the authors.

Introduction
Line 7: Put “.” After al in Caley et al 1996
Lines 10-11: size-dependent?
Lines 14-17: From In this study – it seems that this sentence can be merged to the end paragraph of the Introduction
Line 21: Suggest replacing “among species such as” with “in”
Lines 25-26: Suggest placing “using data for Caribbean octocorals” at the end of the sentence, and remove “,” after octocorals.
Lines 25-27: This seems to be similar thought with Lines 14-17 and aims of study at the end of Introduction. I suggest moving this or merged to aims of study at the end paragraph of Introduction.
Line 27-29: You mentioned in your M&M Lines 108 to 110 that you did not map single polyp recruits, so this is a bit inconsistent here.
Lines 27-33. This can also be merged to lines 81-87, or place somewhere in Materials and Methods section.
Lines 35-6: Arrange citations by years, oldest to recent.
Line 38: Suggest changing “new” with “more”
Line 39: Suggest adding “through fission and budding” after generated.

Materials and Methods
Line 91: Is “Field data.” a subheading here? If so, I suggest putting subheadings in the rest of the parts of M&M for consistency.
Line 92: Remove space in obtained
Lines 94-95: Suggest “The octocoral communities of these sites were almost similar as have been described in Tsounis et al. 2018.”
Lines 95-97: Suggest removing “(“ before Previous and “)” after area.
Line 101: What is the reason for combining recruit numbers for both sites? Have you tried the analyses using each site? As mentioned in Tsounis et al. 2018, Grootpan Bay (identified as East Cabritte, Lasker et al. 2020b) is more exposed with lower sedimentation rates than Europa Bay. So, this site variations might yield a different outcome to your simulations.
Line 102: Suggest adding “(Lasker et al. 2020b)” after <5 cm height. Also, Edmunds and Lasker 2019. Regulation of population size of arborescent octocorals on shallow Caribbean reefs. MEPS
Lines 102-103: Can you provide more information about the 2 sites? And why only two transects were conducted at Grootpan Bay and three at Europa Bay?
Line 103: Although already mentioned in Martinez-Quintana and Lasker (2021) how mapping is done, I think still important to put a sentence here how recruits were mapped and located in the succeeding observations.
Lines 105-107: I think this sentence should be placed in Introduction or Discussion.
Lines 110-111. Since they are very slow growers and possibility of decreasing size due to grazing, etc., is it possible that some initially mapped out recruits were from previous year/s? If so, then perhaps better to specify minimum size in here. Like only individuals with at least 0.3 cm height?
Lines 118-120: So, can you specify in here how many data were used then?
Lines 121-124: I think this sentence is more appropriate in the Introduction, as it states why developing transition matrices is important.
Line 129: Suggest removing “,” after as well as
Lines 130- 133: Can you provide which software you use for all your analyses?
Lines 136-137: To correspond with Fig.1, perhaps better to arrange by “increasing, size remain, or decreasing. Same with positive, no growth, or negative
Line 142: Suggest using “this” instead of “that”
Lines 151-153: I think this sentence came out of nowhere, perhaps better to put it in the last section of M&M. Somewhere between lines 164-174.
Line 155: Suggest “stage-structured”

Results
Line 189: Suggest placing “,” after 1 y
Lines 189-191: As it is a repetition of Line 185, I suggest deleting the sentence. If you want to present % of cohort death and surviving recruits at 5 cm height in 5y, then maybe just merge it with Lines 185 to 189. Or in a separate sentence after with something like “Moreover, after 5 y, 98% of the cohort died…”
Lines 194-195: Suggest replacing “, i.e., those” with “with”. Suggest using “height” instead of “colonies”
Line 196: Suggest changing “, those under 0.6 m” with “(<0.6 m)”
Lines 196-197: The phrase “which illustrates…demographic model”, can be moved to discussion.
Line 202: Suggest using “these” instead of “those”
Line 203: Perhaps you can elaborate a bit on what happens to Fig. 4. By looking at the figure we can tell the high variability in growth rates among different size classes.

Discussion
Lines 228-229: Move to last section of M&M?
Line 250: Can you provide citation for this?
Line 258: Did you mean “rates” instead of “relates”?
Line 262: Suggest transferring “,” after from to after 40%
Line 267: Suggest moving “(Martinez-Quintana and Lasker 2021)” at the end of the paragraph.
Line 270: Since we’re talking about size (height) here, may need to add the size (estimate is fine) of the 3 polyps in the citation.
Line 278: Suggest “Plexaura kuna (Lasker et al. 1998),” and “Briareum asbestinum (Evans et al. 2013).”
Lines 279-280: What type of competition did you mean here? Competition with space, nutrients, etc.? Can you specify? and perhaps add actual observation or citation. Maybe Lasker et al. 1998?
Lines 280-281: Maybe not all Octocorals. Suggest specifying gorgonian Octocorals.
Line 282: Did you mean “offer” instead of “afford”?
Line 284: (Lasker et al. 1998; Evans et al. 2013)
Line 285: Suggest “study” instead of “studies”. Or “…reported here and in Gomez et al. (2014),”
Line 289: Suggest changing “to” with “for”
Line 290: Suggest changing “to” with “for”

Recommendations
I have been wondering why the authors did not perform analyses for each genus, and it is important that they have explained the reasons for that in here.
Line 298: Suggest adding “-“ in between species specific.
Line 303: Suggest using “species” instead of “systems”
Line 312: Suggest adding “d” in demonstrate and adding “with” in between colonies and <5 cm.


Tables: I think it’s easier to follow if you combine transition matrices of empirical model and growth model as 1 table, perhaps both in Table 1. Just separate them as columns and as a.) empirical model and b.) growth model.
Figure 4. Can you specify duration in month/year of observation? August 2016 to?

Reviewer 2 ·

Basic reporting

The Document is well written. Some background studies are necessary for the introduction section. The first two paragraphs of the introduction are really confusing, please rewrite them. Please add, somewhere, the relevance of doing this study using gorgonians as a biological model. what is their role in the ecosystem, etc. The hypothesis of the study is not clearly stated. The methods are clear and sufficient. Interesting results but the discussion can be improved.

Experimental design

The scope is correct for the Journal. The study question is relevant but no ecological context is given or the relevance of doing this study using gorgonians as a biological model. what is their role in the ecosystem etc. Methods are correct and can be replicated.

Validity of the findings

The discussion is a bit short and doesn't include any context of their results, are these results expected? what is the relevance of the data in the context of ecological dynamics in coral reefs? especially in the modern biodiversity crisis?

Additional comments

No comment

Annotated reviews are not available for download in order to protect the identity of reviewers who chose to remain anonymous.

·

Basic reporting

I would like to thank the editors for the opportunity to review this manuscript. I truly enjoyed reading it. The topic is really interesting and the results presented here, help in filling important knowledge gaps on population dynamics for common octocoral species in Caribbean Reefs.

There are a few revisions, that in my view, would help in improving what already is a good manuscript. These are summarised below, with specific comments on grammar and wording provided too.

Although comprehensive and very well explained, most of the background provided about the limitations of surveying coral recruits is scarcely referenced. The points stressed are logical to the reader, however, these have been mentioned in myriad works before and should be better referenced. This is also true for parts of the discussion

The manuscript reads well, but the use of lay language at times compromises its quality. One example of this can be found in lines 121-122: “In many cases, it is impractical to follow individuals over the years or decades that are required to characterize the process of “growing up”. Using phrases such as” to fully model the growth.” would not compromise the ability of the journal’s readership to understand the idea authors are conveying.

Experimental design

Research objectives are well defined and are truly relevant to advance our understanding of the population dynamics of octocorals and fill the many knowledge gaps that exist on coral’s population ecology.

More detail on the methods and results should be provided to avoid having the reader go to other published manuscripts for the information. The summary provided here is too succinct and therefore, this manuscript is lacking enough information on the methods (i.e., analysis such as Chi-square presented in the results; depth at which the surveyed transects were laid, etc) and to an extent in the results for the manuscript to stand alone. Some of the assumptions upon which the analysis approach was based should be stated explicitly (see specific comment about line 212).

Validity of the findings

The discussion read really well and addressed some of the most important results linked to the research objectives stated, however, it also disappointingly lacked implications of these results in terms of the changing composition of Caribbean reefs. Discussing the use of the growth model given the difficulties of following cohorts of corals for long periods of time and in contrast with empirical models would also enrich the discussion and address some of the points raised by the authors in the background/introduction.

Additional comments

Lines 12-14: The second part of this sentence is somewhat confusing, I suggest slightly rewording to: “and different definitions of both recruits and varying methodologies have been used in different studies”

Lines 24-333: This part of the paragraph is repetitive (a summary of what was done is presented at the end of the previous paragraph) and hinders the flow of the Introduction.

Lines 83-85: This sentence is confusing, I suggest rewording it. For example: “Our goals are two-fold: to understand the population dynamics of these small and young colonies and to assess the effects that misidentifying small but older colonies as recruits has on estimates of recruitment rates”

Lines 95-97: Nested brackets

Line 101: Latin names (genus) should be in italics

Lines 146-149: I agree one disadvantage of the empirical model is the relatively low samples from the population used to calculate transition probabilities, however, their distribution within each size class is not a disadvantage in itself, it only exacerbates the low sample size for each probability.

Asymmetric size classes compared to symmetric ones are, for many organisms, a better representation of ontogenic stages key to population rates and it is logical to expect an unbalanced number of individuals in each size class as transition probabilities for each “step” of growth differ.

Line 172: add “, respectively” at the end of the sentence.

Lines 212-214: It would be good to explain the assumptions behind trying to fit the same type of distribution for the growths rates in all size classes and not explore the best fit per size class to then estimate the probabilities

Lines 268, 270, 278: Latin name italicized

Lines 303-309: This caveat should be stated in the methods section, clarifying in line 101 how the authors are certain the polyps -and later on, the small colonies- monitored belonged to those species

---

## Round 0.2 · accepted · Accept

Thank you for this revised version, which is well done. While one reviewer suggested "minor revisions", the comment is format related, and I am happy to move this into production. I look forward to seeing the published version of this work.

Reviewer 1 ·

Basic reporting

The manuscript has greatly improved from the first version. The structure of the sentences and different sections are now more consistent. The authors raise some valid arguments about the scope and limitation of this study.

Experimental design

The study design becomes clearer now with the added information by the authors.

Validity of the findings

Results are much clearer with supporting information in the discussion. The additional discussion also gives a better understanding of the importance of the study such as this.

Additional comments

Clarify whether in situ needs to be italicized or not.

·

Basic reporting

As stated in the first round of revisions, this research topic is really interesting and pertinent, helping fill critical knowledge gaps on population dynamics for common octocoral species in Caribbean Reefs. The manuscript reads really well and the authors have addressed all my previous comments and suggestions regarding the literature referenced and the background.

Experimental design

Research objectives are well defined and are genuinely relevant to advance our understanding of the population dynamics of octocorals and fill the many knowledge gaps in coral’s population ecology. The authors have included more details about their experimental design and data analysis, addressing all my comments/suggestions.

Validity of the findings

The discussion read really well and addressed some of the most important results linked to the research objectives stated within the context of shifting composition within Caribbean coral reefs.

Additional comments

The authors have addressed all suggestions/comments improving what already was a really good manuscript. I again enjoyed reading this research.